# Long-Term Clinical Outcome of First Recurrence Skull Base Meningiomas

**DOI:** 10.3390/jcm9010106

**Published:** 2019-12-31

**Authors:** Yuki Kuranari, Ryota Tamura, Noboru Tsuda, Kenzo Kosugi, Yukina Morimoto, Kazunari Yoshida, Masahiro Toda

**Affiliations:** 1Department of Neurosurgery, Keio University School of Medicine, 35 Shinanomachi, Shinjuku-ku, Tokyo 160-8582, Japan; yuki_kuranari@keio.jp (Y.K.); moltobello-r-610@hotmail.co.jp (R.T.); kensan03977@yahoo.co.jp (K.K.); yukinaxnashiko@yahoo.co.jp (Y.M.); kazrmky@keio.jp (K.Y.); 2Department of Pathology, Keio University School of Medicine, 35 Shinanomachi, Shinjuku-ku, Tokyo 160-8582, Japan; akaishidake@gmail.com

**Keywords:** skull base meningioma, recurrence, atypical, anaplastic, adjuvant radiotherapy

## Abstract

Skull base meningiomas (SBMs) are considered to be less aggressive and have a slower growth rate than non-SBMs. However, SBMs often develop local recurrences after surgical resection. Gross total removal is difficult because SBMs are deep-seated tumors and involve critical neurovascular structures. The treatment strategy for recurrent SBMs remains controversial. The present study aimed to evaluate the long-term clinical course and prognostic factors associated with shorter progression-free survival (PFS) of recurrent SBMs. This retrospective study included 85 recurrent SBMs from 65 patients who underwent surgery from January 2005 to September 2018. Overall survival (OS) and PFS were evaluated, and the associations among shorter PFS and age, sex, tumor size, lesions, World Health Organization (WHO) grading, removal rate, and time since prior surgery were analyzed. The median follow-up period for PFS was 68 months. The 2-, 5-, and 10-year PFS rates were 68.0%, 52.8%, and 22.7%, respectively. WHO grade II or III, multiple lesions, and tumor size were significantly associated with shorter PFS (*p* < 0.0001, *p* = 0.030, and *p* = 0.173, respectively). Although, radiotherapy did not improve PFS and OS for overall patients, PFS of the patients with subtotal and partial removal for WHO grade II SBMs was significantly improved by the radiotherapy. Multivariate analysis identified WHO grade II or III and multiple lesions as independent prognostic factors for shorter PFS (*p* < 0.0001 and *p* = 0.040, respectively). It is essential to estimate the risks associated with shorter PFS for patients with recurrent SBMs to aid in the development of appropriate postoperative strategies.

## 1. Introduction

The growth rate of meningiomas is known to differ depending on their location [1]. Skull base meningiomas (SBMs) typically grow slower and are more likely to show a lower MIB-1 index than non-SBMs [2,3,4]. The WHO grade I subpopulation comprises a vast majority of SBMs [1,2,4,5]. Recurrent SBMs have a relatively long overall survival (OS) [6]. However, compared with non-SBMs, SBMs often develop local recurrences after surgical resection [7,8], which leads to the gradual deterioration of patients’ quality of life. Gross total removal (GTR) is difficult because SBMs are deep-seated, and surrounded by critical vascular structures and cranial nerves [1,7,9,10,11].

Although, other treatment strategies, including chemo-radiotherapy, have been previously used for patients with recurrent and progressive meningiomas [12,13,14], chemotherapies, including sunitinib [13], bevacizumab [14], and everolimus [14], have shown limited efficacy. Recently, several studies have demonstrated the efficacy of adjuvant radiotherapy for high-grade meningiomas [15,16,17] and SBMs [18,19] after initial surgical resection. Various forms of adjuvant radiotherapy, including fractionated radiotherapy and stereotactic radiosurgery, have been found to improve local control [16,17,19]. However, the efficacy of adjuvant radiotherapy for the recurrent stage of SBMs remains unclear [16,17]. Few studies have specifically investigated prognostic factors of recurrent SBMs. Therefore, the treatment strategy for recurrent SBMs remains controversial.

The present study aimed to evaluate the long-term clinical courses and prognostic factors associated with shorter progression-free survival (PFS) and OS of recurrent SBMs for the development of appropriate treatment strategies for recurrent SBMs.

## 2. Materials and Methods

This study was approved by the Institutional Review Board of Keio University (Reference number: 20,050,002), and informed consent was obtained from all patients.

### 2.1. Study Population

A total of 783 cases of patients who were surgically treated with meningiomas were screened at our hospital from January 2005 to September 2018. Among those, this retrospective study included 85 cases of recurrent SBMs from 65 patients (Figure 1). SBMs were defined, as described previously [20].

The exclusion criteria of this study were as follows: (1) Patients who underwent an optic nerve or orbital decompressive operation without tumor resection (*n =* 4); and (2) patients with a small surgical specimen which made a definitive diagnosis difficult (*n =* 1).

Surgical data were retrieved from operative reports, and information on tumor histology was obtained from pathology reports. The data regarding the length of hospital stay and postoperative complications were also obtained. All other information was collected from hospital paper and electronic medical charts. Gadolinium (Gd)-enhanced T1-weighted magnetic resonance imaging (MRI) was used to evaluate tumor size and location. Tumor recurrence was defined as follows: (1) For patients with complete resection, the appearance of new lesions at the prior surgical site; (2) for patients with incomplete resection, residual tumor growth (>2 mm/year); and (3) appearance of the disseminated lesion.

Tumor size was analyzed using the longest diameter, as described previously [21,22]. For multiple meningiomas, tumor size was defined as the longest diameter of the largest one [23]. The removal rate was validated using postoperative routine head computed tomography at 7 days after the operation. Routine postoperative MRI was performed every 6–12 months.

The Simpson grading scale was also used to evaluate the removal rate [24]. The removal rate was categorized as GTR (Simpson grade I and II), subtotal removal (STR) (Simpson grade III), and partial removal (PR) (Simpson grade IV), as previously described [1,10]. In patients with a planned two-stage surgery, the total removal rate was used for the analysis.

PFS was defined as the time that elapsed between treatment initiation and tumor progression. OS was defined as the time from treatment initiation to death. Patients still alive at the last visit were censored as the date of the last follow-up.

### 2.2. Statistical Analysis

Students’ *t*-test was used to compare the length of hospital stay and complication rate among reoperations. PFS and OS were estimated using the Kaplan–Meier method and the log-rank test. We performed univariate analysis with Cox regression models to investigate the poor prognostic factors of recurrent SBMs after surgical resection. Variables with a *p*-value of < 0.2 were included in a subsequent multivariate analysis. A *p*-value of < 0.05 was considered to be statistically significant. Analyses were performed using JMP (version 14.0, SAS Institute Inc., Cary, North Carolina, NC, USA).

## 3. Results

### 3.1. Patient Characteristics

The characteristics of the 65 patients with recurrent SBMs (male: 12, female: 53) have been summarized in Table 1. The median follow-up period was 68 months (range, 0–130 months). The median age at reoperation was 59 years (range, 28–82 years). Among the 65 patients, WHO grade I meningiomas were observed in 52 patients, grade II meningiomas in 12, and grade III meningioma in 1. The median size of the recurrent tumor was 37 mm (range, 4–70 mm). Multiple meningiomas were observed in 10 patients. Most of the recurrent SBMs were located in the middle or posterior cranial fossa (middle: 28, posterior: 27). Only 1 patient showed meningiomatosis with extensive dural thickness at the time of recurrence.

### 3.2. Operation Characteristics

Table 2 summarizes the operation characteristics at the time of recurrence; 85 operations (first reoperation: 53, second reoperation: 23, third reoperation: 9) were analyzed. Seventy-two craniotomies, 12 endoscopic endonasal approaches, and one combined approach (craniotomy + endoscopic endonasal approach) were performed. Four operations were planned two-stage surgeries. Surgical approaches for each craniotomy are summarized in Table 3. The frontotemporal approach and anterior transpetrosal approach [25] were frequently used. GTR (Simpson grade I and II), STR (Simpson grade III), and PR (Simpson grade IV) were achieved in 27, 33, and 21 operations, respectively (Table 3). Forty-nine operations were performed after both symptomatic and radiographical progression and 32 operations after radiological progression alone. Visual impairment was observed before 20 reoperations and proptosis and facial numbness before 7 reoperations. Overall, 43 WHO grade I meningiomas and 10 grade II meningiomas were observed in the first reoperation; 14 grade I, 8 grade II, and 1 grade III in the second reoperation; and 4 grade I and 5 grade II in the third reoperation.

### 3.3. Length of Hospital Stay and Postoperative Complications

Length of hospital stay (days) and postoperative complications are summarized in Table 4. Length of hospital stay was 22.1, 19.6, and 45.9 days for each reoperation, respectively. Length of hospital stay at the third reoperation was significantly longer than that at the first or second reoperation (*p* = 0.0059 and *p* = 0.0066, respectively).

Forty-eight complications were observed after all operations for 32 patients (Table 4). Of 53 patients who underwent first reoperation, 20 patients had complications. In the present study, new cranial nerve deficits and infections were the major complications for first recurrence SBMs. Eight out of 23 patients and 4 out of 9 patients had postoperative complications at the second and third reoperations, respectively. There were no differences in the complication rate among the reoperations (*p* = 0.2455 and *p* = 0.5174, respectively).

### 3.4. Tumor Histology, Removal Rate, and Postoperative Radiotherapy of Each Reoperation

Tumor histology, removal rate, and postoperative radiotherapy at each reoperation are summarized in Table 5. In our institution, patients underwent adjuvant radiotherapy, based on tumor WHO grade, location, and removal rate. Adjuvant radiotherapy was performed for the most patients with WHO grade II SBMs who could not achieve GTR. SRS/SRT and IMRT/3D CRT were used as the adjuvant radiotherapy. In contrast, in patients with WHO grade I SBMs, adjuvant radiotherapy was not routinely performed. However, 9 patients with progressive WHO grade I SBMs who could not achieve GTR received adjuvant radiotherapy.

### 3.5. OS and PFS

A total of 9 patients died within the follow-up period. PFS rates at 2, 5, and 10 years after the first reoperation were 68.0%, 52.8%, and 22.7%, respectively (Figure 2). OS rates at 2, 5, and 10 years after the first reoperation were 94.3%, 86.4%, and 71.7%, respectively (Figure 2).

PFS, after the first to second reoperation, was significantly longer than that after the second to third reoperation and from the third reoperation (*p* = 0.0053) (Figure 3A). The OS of the patients with one reoperation was significantly longer than that of the patients with two and three reoperations (*p* = 0.0005) (Figure 3B). The removal rate at the first reoperation was not associated with the longer PFS and OS (Figure 3C and 3D) (*p* = 0.5918 and *p* = 0.7160, respectively).

The effect of adjuvant radiotherapy was also evaluated for recurrent SBMs (Figure 4). Radiotherapy did not improve PFS and OS for overall patients (Figure 4A and 4B) (*p* = 0.8635 and *p* = 0.4527, respectively). In patients with WHO grade I SBMs, adjuvant radiotherapy did not improve the PFS and OS (Figure 4C and 4D) (*p* = 0.1233, and *p* = 0.5685, respectively). In contrast, in patients with WHO grade II SBMs, adjuvant radiotherapy significantly improved PFS (Figure 4E) (*p* = 0.0012).

### 3.6. Univariate and Multivariate Analyses

To investigate the poor prognostic factors of recurrent SBMs after surgical resection, we performed univariate analysis with Cox regression models for the following variables: Age, sex, neurological progression (present versus not present), time since prior surgery (≥10 years versus <10 years), tumor size (≥60 mm versus <60 mm), lesions (multiple versus single), WHO grade (II or III versus I), and removal rate (PR versus GTR or STR) (Table 6). Among these variables, WHO grading (II or III), lesions (multiple), and tumor size (≥60 mm) were associated with shorter PFS and were included in the subsequent multivariate analysis (WHO grading, HR: 6.686, 95% CI = 2.685–16.647, *p* < 0.0001; lesions, HR: 2.810, 95% CI = 1.104–7.156, *p* = 0.030; tumor size, HR: 2.203, 95% CI = 0.756–4.759, *p* = 0.173).

Multivariate analysis was performed using WHO grading, lesions, and tumor size. The analysis showed that WHO grading (II or III) and lesions (multiple) were independent predictors of poor prognosis (WHO grading, HR: 7.031, 95% CI = 2.739–18.050, *p* < 0.0001; lesions, HR: 2.774, 95% CI = 1.049–7.336, *p* = 0.040).

## 4. Discussion

Several studies have described prognostic factors for primary meningiomas. In a study of 582 patients with primary meningiomas, Amey et al. demonstrated that skull base location, WHO grade II, and Simpson grade III–V resection were independent factors associated with unfavorable outcomes [8]. Similarly, removal rate and WHO grading were reported as prognostic factors for meningiomas in other reports using primary meningiomas [2,26].

Some studies have specifically investigated the difference in prognosis between SBMs and non-SBMs [1,2,3,8,27]. Although SBMs showed less invasive characteristics, such as lower MIB-1 index [1], the median time to recurrence was shorter in SBMs than in non-SBMs [1,8,27]. The difficulty of surgical treatment may be the cause of this discrepancy. GTR is difficult because SBMs are deep-seated and surrounded by critical vascular structures and cranial nerves [1,7,9,10,11]. Kira et al. [10] demonstrated that the proportion of patients with STR was higher in SBMs than in convexity and falx meningiomas, and the removal rate was the major prognostic factor for SBMs. Yu et al. [28] also recommended complete resection to achieve longer PFS on the basis of their clinical experience with 28 patients with atypical SBMs. Nanda et al. [29] demonstrated a significant association between the Simpson grading system and recurrence rates for both skull base meningiomas and convexity meningiomas. However, in the present study, although the removal rate (including Simpson grading scale) was reported as the strongest prognostic factor for primary meningiomas, it was not associated with shorter PFS of recurrent SBMs.

Although, the removal rate was reported to be associated with the prognosis of SBMs, aggressive removal must be carefully discussed. Aggressive removal was known to increase postoperative complications for SBMs. In the present study, new cranial nerve deficits and infections were the major complications for recurrent SBMs. The previous study demonstrated that the complication rate of recurrent SBMs is higher than that of other meningiomas [6].

In general, higher WHO grading has been associated with the poor prognosis of meningiomas [30]. According to a previous analysis of primary SBMs, 16 of 358 patients had WHO grade II meningiomas [10]. In SBMs, WHO grade II meningiomas were more common at the medial and lateral sphenoid ridge [1]. Atypical SBMs show poor clinical outcomes, but the literature regarding atypical SBMs is quite limited [28]. In the present study, most of the grade II meningiomas originated from the middle cranial fossa. We found that WHO grading was strongly associated with shorter PFS of recurrent SBMs. Although atypical and anaplastic meningiomas have been rarely observed in SBMs previously [4,5], atypical and anaplastic histological features have found to be important prognostic factors.

In other reports, tumor size, location, and bone- and venous sinus infiltration were reported as prognostic factors for SBMs [19,31]. Although these factors were not associated with the poor prognosis of recurrent SBMs, multiple lesions and/or dissemination were significant prognostic factors in the present study. In a previous study on disseminated meningiomas, tumor origins and WHO grading were not associated with dissemination [32]. The pathogenesis of leptomeningeal dissemination is still unknown.

In the recent decade, the efficacy of adjuvant radiotherapy has been demonstrated for high-grade meningiomas [15,16,17]. Hilary et al. demonstrated that adjuvant radiotherapy for meningiomas resected with Simpson grade I–III improved local control rates [16]. The prognosis of primary anaplastic meningiomas has been shown to be improved by adjuvant radiotherapy; however, in the subgroup analyses, no significant improvement was observed in either PFS or OS in patients with recurrent meningiomas [17]. In the present study, radiotherapy did not improve PFS and OS for overall patients. However, PFS of the patients with subtotal and partial removal for WHO grade II SBMs was significantly improved by the radiotherapy. Therefore, further development of the treatment strategy using radiotherapy is expected. For example, novel treatment strategies for recurrent SBMs, such as brachytherapy [33], may show treatment efficacy.

Although, it is difficult to completely establish a treatment strategy for recurrent SBMs, some diagnostic radiographical modalities may be useful [34,35]. Ching–Chung et al. [35] demonstrated that low apparent coefficient diffusion values were high-risk factors for the progression of SBMs.

The present study had some limitations. The main limitation was the small sample size, given that the prevalence of SBMs is relatively rare [1]. Further analysis with a larger sample size of patients with recurrent SBMs is required to confirm the findings of the present study.

## 5. Conclusions

To date, a few studies with a specific focus on recurrent SBMs have been reported. The present study demonstrated that WHO grading and multiple lesions were associated with shorter PFS of recurrent SBMs. PFS of patients with subtotal and partial removal for WHO grade II SBMs was significantly improved by the radiotherapy.

## Figures and Tables

**Figure 1 jcm-09-00106-f001:**
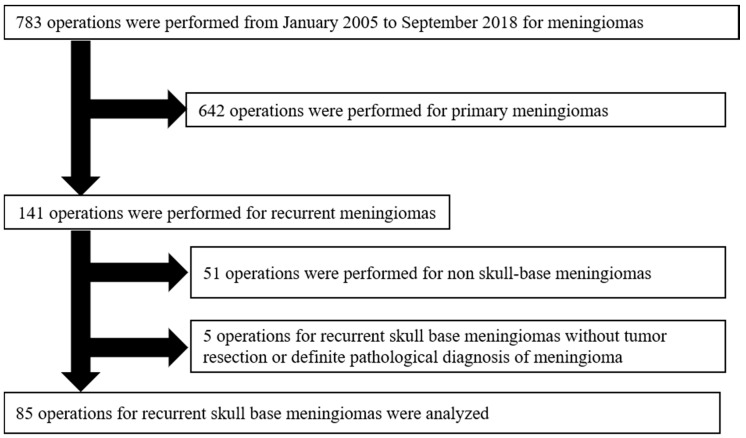
Patient selection in the study. A total of 783 surgically treated cases with meningiomas were screened. Among those, this retrospective study included 85 recurrent SBMs from 65 patients. The exclusion criteria were as follows: (1) Patients who underwent an optic nerve or orbital decompressive operation without tumor resection (*n =* 4), and (2) patients with a small surgical specimen, for which it was difficult to make a definitive diagnosis (*n =* 1).

**Figure 2 jcm-09-00106-f002:**
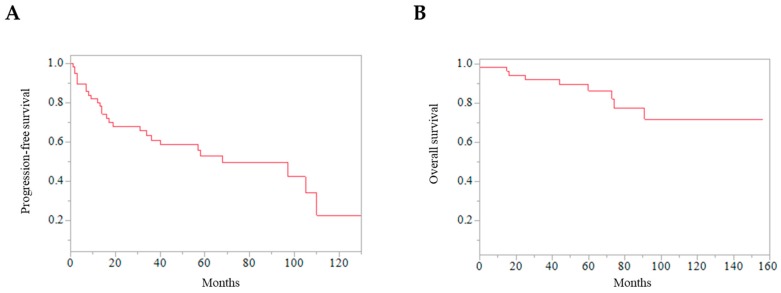
Kaplan–Meier curves of all the patients. (**A**) Progression-free survival after reoperation of skull base meningiomas. (**B**) Overall survival after reoperation of skull base meningiomas.

**Figure 3 jcm-09-00106-f003:**
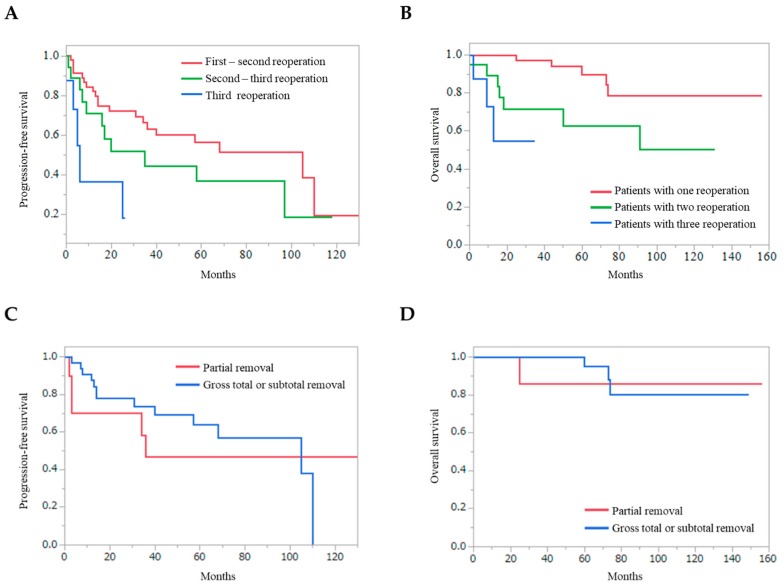
Kaplan–Meier curves, that are dependent on repeat reoperations and removal rate. (**A**) Progression-free survival from the first to second reoperation was significantly longer than that from the second to third reoperation and that from the third or later reoperations (*p* = 0.0053). (**B**) Overall survival of the patients with one recurrence was significantly longer than that of the patients with two and more than three recurrences (*p* = 0.0005). Progression-free survival (**C**) and overall survival (**D**) from the first reoperation depending on the removal rate are shown.

**Figure 4 jcm-09-00106-f004:**
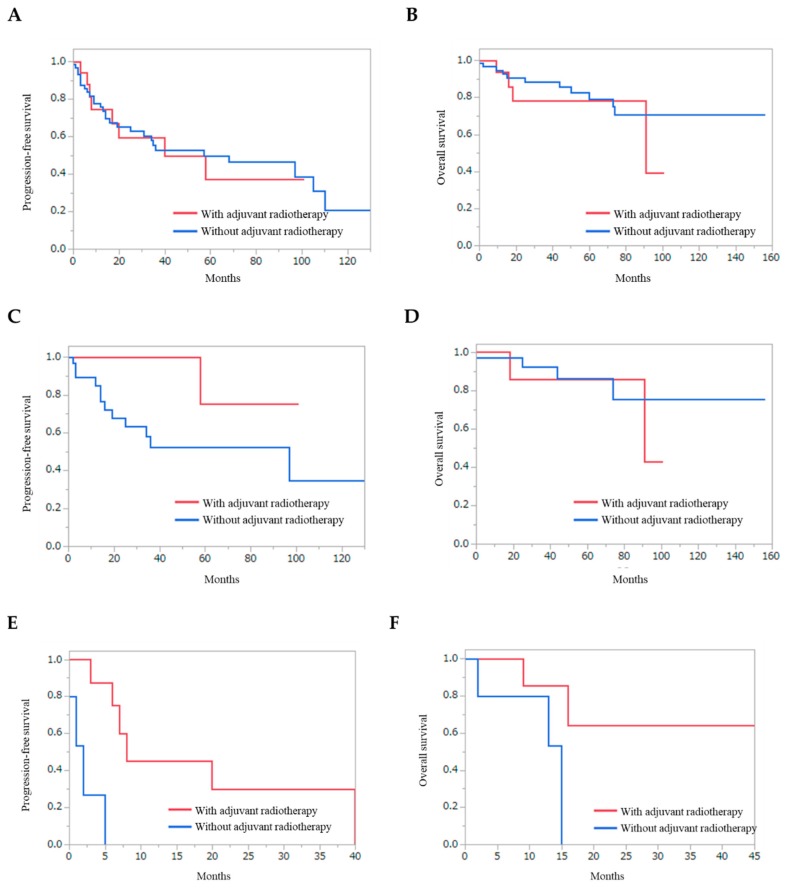
Kaplan–Meier curves of the patients depending on the adjuvant radiotherapy. Progression-free survival (**A**) and overall survival (**B**) of overall patients with recurrent skull base meningiomas. Progression-free survival (**C**) and overall survival (**D**) of the patients with subtotal and partial removal of WHO grade I skull base meningiomas. Progression-free survival (**E**) and overall survival (**F**) of the patients with subtotal and partial removal of WHO grade II skull base meningiomas. Adjuvant radiotherapy significantly improved PFS for patients with WHO grade II skull base meningiomas (Figure 4E) (*p* = 0.0012).

**Table 1 jcm-09-00106-t001:** Patient characteristics (*N =* 65).

Parameter	Number (%)
**Sex**	
Male	12 (17.4)
Female	53 (82.6)
**Time since prior surgery**	
< 10 years	48 (73.8)
≥ 10 years	17 (26.2)
**WHO grading**	
I	52 (80.0)
II	12 (18.5)
III	1 (1.5)
**Tumor size (mm)**	37 (4-70)
Diameter ≥ 60mm	11 (16.9)
Diameter ≥ 30mm, < 60mm	32 (49.2)
Diameter < 30mm	22 (33.9)
**Lesions**	
Multiple	10 (15.4)
Single	55 (84.6)
**Location**	
*Anterior cranial fossa*	9 (13.9)
Olfactory groove	2 (3.1)
Tuberculum sellae	7 (10.8)
*Middle cranial fossa*	28 (43.1)
Sphenoid wing	4 (6.2)
Anterior clinoid process	5 (7.7)
Cavernous sinus	7 (10.8)
Floor	7 (10.8)
Optic canal and orbit	3 (4.6)
Meckel’s cave	1 (1.5)
Spheno-orbital	1 (1.5)
*Posterior cranial fossa*	27 (41.5)
Petroclival	9 (13.8)
Spheno-petroclival	2 (3.1)
Petrosal bone	7 (10.8)
Jugular foramen	1 (1.5)
Tentorial	5 (7.7)
Foramen magnum	3 (4.6)
*Meningiomatosis*	1 (1.5)

WHO, World Health Organization.

**Table 2 jcm-09-00106-t002:** Reoperation characteristics (*N =* 85).

Parameter	Number (%)
**Operation**	85
First reoperation	53 (63.3)
Second reoperation	23 (26.7)
Third reoperation	9 (10.0)
**Two-staged surgery**	4 (4.7)
**Symptomatic progression**	49 (60.5)
Visual impairment	20 (24.7)
Proptosis	7 (8.6)
Facial numbness	7 (8.6)
Headache	3 (3.7)
Gait disturbance	3 (3.7)
Cognitive changes	2 (2.5)
Hearing disturbance	2 (2.5)
Other symptoms	5 (6.2)
**Radiological progression**	32 (39.5)

**Table 3 jcm-09-00106-t003:** Detailed information of surgical approach and removal rate (*N =* 85).

Variable	First Reope	Second Reope	Third Reope	Total
**Number of operations**	53	23	9	85
*Craniotomy*	47	19	6	72
Frontotemporal approach	14	8	4	26
Orbitozygomatic approach	5	2	0	7
Anterior transpetrosal approach	8	3	0	11
Combined transpetrosal approach	6	2	0	8
Posterior transpetrosal approach	1	0	0	1
Lateral suboccipital approach	3	0	0	3
Transcondylar fossa approach	3	1	0	4
Other approach	7	3	2	12
*Endonasal endoscopic surgery*	6	3	3	12
*Combined approach (Craniotomy+* *Endonasal endoscopic surgery)*	0	1	0	1
**^#^ Removal rate**				81
Gross total removal (Simpson grade I and II)	21	5	1	27
Subtotal removal (Simpson grade III)	19	11	3	33
Partial removal (Simpson grade IV)	10	6	5	21

Reope, reoperation. ^#^ In patients with planned two-stage surgery, the total removal rate was used for the analysis.

**Table 4 jcm-09-00106-t004:** Length of hospital stay and complications for each reoperation (*N =* 85).

Variable	First Reope	Second Reope	Third Reope	Total
**Number of operations**	53	23	9	85
**Mean length of hospital stay (days)**	22.1	19.6	45.9	
**Number of patients with complications** **(Total number of complications)**	20 (35)	8 (9)	4 (4)	32 (48)
**Complications**	35	9	4	48
*New cranial nerve deficits*	15	2	0	17
CN II	3	0	0	3
CN III	1	0	0	1
CN IV	2	0	0	2
CN V	1	0	0	1
CN VI	3	0	0	3
CN VII	5	2	0	7
*Other neurological deficits*	4	2	0	6
Diabetes insipidus	2	2	0	4
Hemiparesis	1	0	0	1
Dysphagia	1	0	0	1
*Postoperative hematoma*	1	1	0	2
*Cerebral infarction*	1	0	0	1
*Cerebral contusion*	0	1	0	1
*Cerebral edema*	1	2	0	3
*Vascular injuries*	0	0	1	1
*CSF leakage*	2	0	0	2
*Hydrocephalus*	1	0	1	2
*Infection*	8	0	1	9
Meningitis	4	0	0	4
Urinary tract infection	1	0	1	2
Other infection	3	0	0	3
*Other complications*	2	1	1	4

Reope, reoperation; CN, cranial nerve.

**Table 5 jcm-09-00106-t005:** Detailed information of tumor histology, removal rate and postoperative radiotherapy at each reoperation (*N =* 85).

Variable		First Reope	Second Reope	Third Reope	Total
**Number of operations**		53	23	9	85
*Tumor histology*	*^#^ Removal rate*				
WHO grade I	GTR	15	2	0	17
	STR	16	7	0	23
	PR	10	4	4	18
WHO grade II	GTR	6	3	1	10
	STR	3	4	3	10
	PR	0	1	1	2
WHO grade III	PR	0	1	0	1
*Adjuvant radiotherapy*		9	6	2	17
SRS/SRT		8	3	2	13
IMRT/3D CRT		1	3	0	4
WHO grade I	STR	4	2	0	6
	PR	2	0	1	3
WHO grade II	STR	3	3	1	7
	PR	0	1	0	1

Reope, reoperation; WHO, World Health Organization; GTR, gross total removal; STR, subtotal removal; PR, partial removal; SRS, stereotactic radiosurgery; SRT, stereotactic radiotherapy; IMRT, intensity modulated radiation therapy; 3D CRT, three dimensional conformal radiotherapy. ^#^ The removal rate was categorized as GTR (Simpson grade I and II), subtotal removal (STR) (Simpson grade III), and partial removal (PR) (Simpson grade IV).

**Table 6 jcm-09-00106-t006:** Univariate and Multivariate Analysis.

Variables	Univariate Analysis			Multivariate Analysis		
	HR	(95% CI)	*p* Value	HR	(95% CI)	*p* Value
Age	1.004	(0.976–1.033)	0.785			
Sex (Female)	1.486	(0.511–4.322)	0.468			
Neurological progression (Present)	1.672	(0.750–3.731)	0.209			
Time since prior surgery (≥10 years)	0.562	(0.193–1.636)	0.291			
Tumor size (≥60 mm)	1.896	(0.756–4.759)	0.173	2.203	(0.862–5.630)	0.099
**Lesion (Multiple)**	2.810	(1.104–7.156)	0.030	2.774	(1.049–7.336)	**0.040**
**WHO grade (II or III)**	6.686	(2.685–16.647)	<0.001	7.031	(2.739–18.050)	**<0.0001**
Removal rate (PR)	0.984	(0.396–2.443)	0.972			

HR, hazard ratio; CI, confidence interval; WHO, World Health Organization; PR, partial removal. Boldface type indicate statistical significance.

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
