# Peer review of "Long-Term Clinical Outcome of First Recurrence Skull Base Meningiomas"

_jcm, 2019, doi:10.3390/jcm9010106_

Round 1
Reviewer 1 Report
Recurrent skull base meningiomas are not a novel topic, but remain a clinical concern. The authors’ analysis on 65 patients with recurrent skull base meningiomas, identified WHO grade II or III and multiple lesions as independent prognostic factors for shorter PFS. These are not modifiable factors, intrinsic to the tumor.
The role of radiotherapy on recurrent meningiomas remains controversial.
The data on the patients who underwent radiotherapy are not clearly presented. The patients underwent RT after first surgery, after first recurrence or after second surgery, based on histology, on extension of removal, or both? More detailed information would be necessary, to interpret the result reported (“Radiotherapy did not improve the PFS and OS in the patients with recurrent SBMs”).
Author Response
We are very grateful to you for the insightful comments and suggestions, which have undoubtedly helped us to improve our manuscript immensely. As indicated in the responses below, we have taken all the comments and suggestions into account when generating the revised version.
As the reviewer indicated, the role of radiotherapy on recurrent meningiomas remains controversial. In our institution, postoperative radiotherapy was performed based on tumor histology, location, and removal rate. We have added a new Table 5, which includes more detailed information on postoperative radiotherapy. In addition, subgroup analysis was performed to interpret the results of radiotherapy for recurrent SBMs. Although radiotherapy did not improve PFS and OS for overall patients, PFS of the patients with subtotal and partial removal for WHO grade II SBMs was significantly improved by the radiotherapy. Therefore, further development of the treatment strategy using radiotherapy is expected. We added a new Figure 4 in the revised manuscript.
Reviewer 2 Report
The authors have assessed long-term clinical outcomes in 65 patients with recurrent skull base meningiomas who under 85 operations over a 14-year period. The results are focused on long-term survival and mortality. Although an important topic, this manuscript is seriously lacking in methodology, including what transcranial approach was used, what constitutes GTR, STR and partial removal, and how they used the Simpson grading scale (there does not appear to be any Simpson data in the results). Presumably, most resections were Simpson Grade 2 or higher.
Most importantly, there is no data on surgical complications and length of hospital stay. Without detailed data on surgical complications (cranial nerve deficits, other neurological deficits, vascular injuries, postoperative hematomas, CSF leaks, meningitis, etc) and length of stay. Finally, it is not clear in the results how many patients received radiation (and what type of radiation) after their reoperation. Also was RT effective in tumor control?
Author Response
We are very grateful to you for the insightful comments and suggestions, which have undoubtedly helped us to improve our manuscript immensely. As indicated in the responses below, we have taken all the comments and suggestions into account when generating the revised version.
Thank you very much for your favorite comments. We added the information about methodology. Surgical approaches, removal rate, and Simpson grading scale were added into Table 2 and the new Table 5. As previously described, the removal rate was categorized as GTR (Simpson grade I and II), subtotal removal (STR) (Simpson grade III), and partial removal (PR) (Simpson grade IV) [Mansouri A. J Neurosurg. 2016; Voss K.M. J Neurooncol. 2017]. As the reviewer indicated, most resections were Simpson grading scale two or higher.
In addition, we added a new Table 4, which summarizes the data on surgical complications and length of hospital stay for each reoperation. Finally, a new Table 5 and Figure 4 were also added to show the detailed information of postoperative radiotherapy. SRS/SRT and IMRT/3D CRT were used as adjuvant radiotherapy. Subgroup analysis was performed to interpret the results of radiotherapy for recurrent skull base meningiomas. Although radiotherapy did not improve PFS and OS for overall patients, PFS of the patients with subtotal and partial removal for WHO grade II SBMs was significantly improved by the radiotherapy. Therefore, further development of the treatment strategy using radiotherapy is expected.